# Evaluation of Dental Root Development Regarding Maxillary Canine Eruption Status after Secondary Alveolar Bone Grafting in Patients with Cleft Lip and Palate

**DOI:** 10.3390/diagnostics13091642

**Published:** 2023-05-06

**Authors:** Melissa A. Ferguson, Sercan Akyalcin, Hugo Campos, Abigail Gliksten, Kadriye Hargett, Stephanie Yang, James MacLaine

**Affiliations:** 1Developmental Biology, Harvard School of Dental Medicine, Boston, MA 02115, USA; 2Evidence-Based Health Care Program, University of Oxford, Oxford OX1 3PJ, UK; 3Department of Dentistry, Boston Children’s Hospital, Boston, MA 02115, USA

**Keywords:** cleft lip and palate, secondary alveolar bone graft (SABG), cone beam computer tomography (CBCT), maxillary canine eruption

## Abstract

In children born with cleft lip and palate, the timing of the secondary alveolar bone graft (SABG) is crucial to its success; this involves estimating the eruption of the permanent maxillary canine. Altered dental eruption in this patient group gives impetus to the identification of dental developmental factors concerning maxillary canine eruption, which may steer the clinical decision of SABG timing. Records of over nine hundred patients who received SABG with pre- and post-operative cone beam computed tomography (CBCT) scans were analyzed for inclusion and divided into two groups (erupting or non-erupting canine after SABG). Roots of the maxillary canines and premolars were segmented from the cementoenamel junction then linear and volumetric measurements were performed. The pre- and post-operative root length and volume differences were calculated and compared statistically using independent sample tests and paired *t*-tests. No statistically significant differences were found in the volume change (%), or reciprocal of mean root length in the erupted and unerupted groups in the canine, first premolar, or second premolar roots except for an association between the post-operative dental root length of the canine and the maxillary canine eruption status. Therefore, assessment of root development from pre-treatment CBCT scans was not deemed worthy from a diagnostic perspective.

## 1. Introduction

In the United States, cleft lip and palate is the most common birth defect with approximately 6800 babies born affected each year [1]. Cleft lip and palate falls under the umbrella of oral clefts [2] and can involve isolated cleft lip, both cleft lip and cleft palate or isolated cleft palate [3]. Alveolar clefts are present in 75% of patients with cleft lip or cleft lip and palate [4]. A complete alveolar cleft will transverse the alveolus [5]. The alveolar cleft will remain present in the alveolus following the early surgical procedures patients with cleft lip and palate undergo for lip and palate repair which will necessitate for the alveolar cleft to be addressed at a later age with alveolar bone grafting [5]. Alveolar bone grafts are necessary in the management of cleft lip and palate patients to repair the alveolar bony defect, restore the arch continuity and improve the anatomic contour for orthodontic or implant treatment, close oroantral fistulas that may be present, improve the anatomical morphology of the alar base, and provide periodontal and boney support for the eruption of the lateral incisors and canine teeth adjacent to the cleft [4,5,6,7,8].

Alveolar bone grafting is a technique used for bone atrophy in order to restore an adequate bone volume: autologous, heterologous and xenograft are possible materials used in this technique [9,10,11]. Alveolar bone grafting can be performed at different stages in children with cleft lip and palate, with the classification based on the dentition of the patient [4]. Primary alveolar bone grafting is completed in the deciduous dental developmental stage [4,5] and is sequenced prior to the eruption of the deciduous teeth [12]. Primary alveolar bone grafting can be performed simultaneously with the lip repair in infancy [13], between 0 and 24 months of age [12]. Primary alveolar bone grafting can prevent maxillary collapse, support the alar base, and improve the ability of the child to eat [14]. Disadvantages of primary alveolar bone grafting include an adverse effect on lateral maxillary growth [15,16] and a lack of maintenance of the vertical dimension of the bone [14]. Long-term results from patients who underwent primary alveolar bone grafting showed midface deficiency from a lack of normal development and growth after the procedure [17]. Secondary alveolar bone grafts (SABG) are performed during the mixed dentition [4,5,7]. SABG has the advantage of only a minor disruption to facial growth as facial growth is completed mainly by the time of this grafting procedure [14,15]. There is also improved stability of the bony environment and improved periodontal health of the permanent maxillary canines due to their imminent eruption through the alveolar bone graft [14], which stimulates the formation of additional alveolar bone [18]. Tertiary alveolar bone grafts are performed after mixed dentition [4,5], which will avoid interfering with facial growth, similar to SABG [14]. However, tertiary alveolar bone grafts have the disadvantage that the teeth adjacent to the cleft may be compromised due to a lack of bone support [14], and this bone graft lacks eruption stress which functionally loads the graft [5]. SABG is the gold standard of treatment for grafting alveolar clefts and has been well-documented in the literature to be advantageous over primary and tertiary bone grafts [1,4].

There are different types of grafting materials that can be used for secondary alveolar bone grafts. The gold standard for grafting in SABG is iliac cancellous bone [19,20].The anterior iliac crest is the most common grafting donor site [20]. Iliac cancellous bone provides abundant cancellous bone which contains osteogenic and pluripotent cells and allows for revascularization to happen quickly [20]. However, the donor site has post-operative pain and scarring associated with the harvest [19]. Calvarial cortical bone has also been used as a grafting material in SABG due to the close proximity of the donor and recipient site with similar embryonic origins [19]. SABG performed with iliac cancellous grafts have a success rate of 89.9% compared with 63% in calvarial bone grafts [21]. Alternatively, tibial bone can be used as a grafting material as it is also a cancellous bone [20], but there is a risk of damage to epiphyseal cartilage during the donor harvest in these growing patients [19]. Due to morbidity at donor sites, bone substitutes have been investigated. Bone substitutes include allogenic freeze-dried bone and recombinant human bone morphogenetic protein, both of which lack long-term safety data in patients with cleft lip and palate [19].

Timing of the SABG procedure is essential and should be performed before the onset of the permanent maxillary canine eruption. Grafting before the permanent maxillary canine eruption will allow the canine to erupt into the graft, which will functionally load the graft [2]. A study by Bergland et al., which investigated SABG outcomes in patients aged 8–18 years old, found that the best results were in patients who received bone grafting before the maxillary canine eruption. Normal height of the interdental septum was achieved in 64% and 37% of patients with SABG before and after the canine eruption, respectively (*p* < 0.05) [18]. In addition, Bergland et al. found orthodontic space closure was more likely in patients who underwent SABG before the canine erupted (90%). The underlying space closure mechanism was due to the mesial drift of the erupting canine into the graft. In contrast, 72% of patients had successful orthodontic space closure and the remaining 28% of patients required prosthetic restorations to close the edentulous space in patients who underwent SABG after the canine eruption [18]. Grafting after the canine erupts is less successful than grafting before the canine erupts because the crown of the erupted canine often ingresses into the alveolar defect, thereby preventing graft adherence. Additionally, the graft will lack the functional load associated with canine eruption stress and the adjacent teeth can have periodontal defects [5]. Conversely, SABG should not be performed too early before the permanent maxillary canine eruption because the bone will resorb if there is no tooth function in that area, requiring additional bone grafting. Spontaneous eruption of the canine after bone grafting is believed to be approximately two years, as per the literature [22].

A systematic review proposes that the optimal timing of SABG is between 8 and 12 years of age [6]. Other studies suggest between 9 and 11 years of age [1] when the permanent maxillary canine root is between one-half [1] and two-thirds developed [15]. This use of chronological age ranges is common in the cleft lip and palate literature highlighting the lack of agreed anatomical and developmental norms local to the alveolar defect which could more precisely inform the decision-making algorithm. Therefore, the clinician should use their best judgment based on the position of the unerupted teeth adjacent to the graft and dental development rather than chronological age [17] to determine the appropriate timing for each patient’s SABG procedure.

Tooth development is altered in children with cleft lip and palate [5]. Delayed permanent tooth formation and eruption in children with cleft lip and palate have been investigated [23]. A systematic review of patients with cleft lip and palate of 36 articles found in 32 articles there was a delay in dental development or tooth eruption [24]. Cleft lip and palate is multifactorial; however, genetic factors such as TGF-alpha, TGF-Beta3 and MSX1 are involved in both the secondary palate development and tooth development [24]. Additionally, the association between altered dental development and the presence of a cleft could be due to the interference of neural crest cell migration [24]. There is also some literature that suggests the altered dental development in patients with cleft lip and palate is the result of insufficient bone in the maxilla or previous surgical interventions [24]. Furthermore, altered and smaller tooth morphology in these patients has been suggested to alter the eruption [24]. Children with non-syndromic cleft lip and palate were found to have dental development delays of up to 1.56 years [5]. This study also found delayed tooth eruption in these children on the non-cleft side [5]. Therefore, it is essential to study dental development concerning the permanent maxillary canine eruption in patients with cleft lip and palate due to altered dental eruption evident in these patients, which can complicate planning the timing of SABG.

Following SABG, our null hypothesis is that there is no association between pre-operative dental root formation and maxillary canine eruption status in patients with cleft lip and palate. Our study aims to improve clinicians’ decision-making algorithm when deciding the timing of SABG by investigating the dental development concerning the eruption status of the permanent maxillary canine.

## 2. Materials and Methods

In our retrospective case–control study, we examined the records of 918 patients with cleft lip and palate that underwent SABG at Boston Children’s Hospital. This study was conducted in accordance with IRB # P00038300 approved by Boston Children’s Hospital on 20 September 2021. The clinical notes were used adjunct to the cone beam computed tomography (CBCT) imaging. The inclusion criteria included an infant diagnosis of a non-syndromic unilateral complete cleft lip and palate, presence of unerupted cleft-side canine pre-graft, and patients with CBCT data available pre- and post-SABG with the post-surgical CBCT not taken more than two years after the SABG procedure.

Exclusion criteria included patients who did not have a pre-surgical CBCT available or a follow-up post-surgical CBCT available within two years of the SABG procedure, a diagnosis of syndromic cleft lip and palate, a diagnosis of an incomplete cleft, the presence of a cleft-side canine that has erupted before the graft, the presence of a prime lateral incisor on the cleft-side necessitating early SABG or a diagnosis of bilateral cleft lip and palate.

Deidentified (No PHI) pre-and post-operative CBCT images of the patients were exported for analysis. The patients were divided into two groups after evaluating the post-surgical CBCT scans: (1) cleft-side permanent maxillary canines showing radiographic evidence of erupting within two years of the SABG procedure or (2) cleft side permanent maxillary canines which remained unerupted within two years of the SABG procedure.

### 2.1. Volumetric Analysis

Density values of different tissues, that is teeth vs. bone, were identified in Hounsfield units (HU) in the axial view and used to segment the maxillary canines and premolars from both groups using the commercial software (InVivo Anatomage version 6.5, San Jose, CA, USA). Upon the completion of the segmentation process, the root area was sectioned from the crowns below the cementoenamel junction (CEJ) to exclude the crowns. Then, volumetric measurements of the maxillary canine and premolar roots on the ipsilateral side with the alveolar cleft were performed on the resultant images (Figure 1). The volume change (%) between the pre- and post-operative CBCTs was calculated for the erupting and non-erupting groups for the canine and premolars.

### 2.2. Root Length Analysis

The root lengths were measured (Figure 2) below the CEJ in the axial view to the tip of the forming root using commercial software (Dolphin, Patterson Dental, Saint Paul, MN, USA). A root length ratio was calculated using the length of the root measured on the CBCT compared with standard root lengths [25] for each respective root: 17.3 mm for the permanent maxillary canine, 12.4 mm for the maxillary first premolar and 14.0 mm for the maxillary second premolar. If a maxillary first premolar had two roots, then the root length was measured from the buccal root.

### 2.3. Statistical Analysis

IBM SPSS Statistics version 28.0.0.0 (190) was used for statistical analysis including the two-tailed independent sample t tests, paired sample t tests and bootstrapping. For the power and sample size calculations, Stata/IC 15.1 was used. A two-tailed independent sample *t*-test was performed on the mean root volume change (%) between erupting and non-erupting groups for the canine and premolars. Two-tailed independent *t*-tests and two-tailed paired sample *t*-tests were used to analyze the data for the reciprocal root length ratios. The level of significance was set at *p* < 0.05.

In the independent sample *t*-test and paired sample *t*-test, a sample of size *n* was drawn from the population and will be referred to as sample *S.* These statistical tests are parametric tests which assume the data is normally distributed. It would be reasonable to assume that the data were normally distributed as the *p*-value from the Shapiro–Wilk test was <0.05; however, exploratory data analysis (EDA) revealed skewness and kurtosis, and thus a sensitivity analysis was conducted by repeating the independent sample *t*-test and paired sample *t*-test with bootstrap. In our bootstrap the sampling distribution was created by resampling observations with replacement from S 2000 times with each resampled set having *m* observations. In contrast to a traditional approach bootstrapping does not assume any underlying distributions of the data [26]. The non-parametric bootstrap test can be used for small sample size studies [26] and is reliable for a sample of eight or greater [11], which our study exceeds.

### 2.4. Sample Size

Post hoc sample size calculations were performed using Stata for the independent sample *t*-test for both the volumetric and root length analysis and paired sample *t*-test for root length analysis. A post hoc was performed as there is currently limited literature on volume to estimate effect size.

## 3. Results

In total, 918 patients with cleft lip and palate that underwent SABG at Boston Children’s Hospital were evaluated for inclusion in the study. After examining the CBCTs and medical records, *N* = 60 patients were eligible to be included in the study which was then divided into 40 in the erupting canine group and 20 in the non-erupting canine group.

### 3.1. Volumetric Analysis

There was no difference (*p* > 0.05) in mean root volume change (%) of the canine between the erupting and non-erupting groups (Table 1) with a power of 12% (12% power means there is a 12% chance of detecting a difference between the population mean and the target when a difference actually exists) and after bootstrapping there was no significant difference (*p* > 0.05). There was no statistically significant difference (*p* > 0.05) in the mean root volume change (%) of the first premolar using the independent sample *t*-test with a 61% power (61% power means there is a 61% chance of detecting a difference between the population mean and the target when a difference exists) between the mean of the erupting and non-erupting (Table 1). Further analysis after bootstrapping had a *p*-value > 0.05, indicating that there was no significant difference in the first premolar mean root volume change (%) between erupting and non-erupting canine groups. There was no difference (*p* > 0.05) in mean root volume change (%) of the second premolar between the erupting and non-erupting groups (Table 1) with a power of 12%.

### 3.2. Root Length Analysis

#### 3.2.1. Independent *t* Tests for Root Length

The root length ratios were calculated as described in the methods section by comparing the length measured on the CBCT to standard root lengths [25]. The reciprocal of mean root length ratio was used in the independent sample *t*-test to try to force normality. After the transformation, the Shapiro–Wilk had a *p*-value > 0.05, indicating the data was normally distributed. However, the descriptive statistics showed the data set is skewed; thus, a sensitivity analysis using non-parametric bootstrap was also carried out.

Comparing the reciprocal of the mean root length ratio of the canine between the erupting and non-erupting groups (Table 2), there was no significant difference pre-operatively (*p* > 0.05) with a power of 5%, and after bootstrapping, there was a similar result (*p* > 0.05); however, the reciprocal of the mean root length ratio of the canine was significantly different (*p* = 0.025) post-operatively (Table 3) with a power of 57% which was similar to the result found after bootstrapping (*p* = 0.035). Comparing the reciprocal of the mean root length ratio of the first premolar between the erupting and non-erupting group, there was no statistically significant difference (*p* > 0.05) pre-operatively (Table 2) with a power of 5% and post-operatively (Table 3) with a power of 30% which were both similar to the bootstrapping results (*p* > 0.05). Comparing the reciprocal of the mean root length ratio of the second premolar between the erupting and non-erupting group, there was no significant difference (*p* > 0.05) pre-operatively (Table 2) with a power of 7% and post-operatively (Table 3) with a power of 13% which were both similar to the bootstrapping results (*p* > 0.05).

#### 3.2.2. Paired *t* Tests for Reciprocal of Mean Root Length Ratios

Paired sample *t*-tests were also performed to compare the reciprocal of the mean root length ratio in the same patient before and after the grafting procedure. The descriptive statistics showed skewness and kurtosis; therefore, a sensitivity analysis using non-parametric bootstrapping was performed. In patients with erupting canines (Table 4), the reciprocal of the mean root length ratio for the canine, first premolar, and second premolar was significantly different (*p* < 0.05) pre-operatively and post-operatively. In patients that had a non-erupting canine (Table 4), the canine and first premolar were significantly different (*p* < 0.05) pre-operatively and post-operatively in the paired *t*-test and in the bootstrapping paired *t*-test. The second premolar in the non-erupting group was found to be significant in the paired sample *t*-test (*p* < 0.05), but the bootstrap paired sample *t*-test *p*-value is >0.05 (*p* = 0.07). However, the bootstrap paired sample *t*-test confidence interval is 2.88 to 8.87 and does not cross zero.

## 4. Discussion

Our study proposed a novel approach in analyzing root formation in CBCTs in cleft lip and palate patients. After analyzing pre- and post-operative CBCTs, we found there was no association between pre-operative dental root formation and maxillary canine eruption status in patients with cleft lip and palate; therefore, the null hypothesis was accepted. We could not identify any statistically significant differences in the mean root volume change (%) in the erupted and the unerupted groups in the canine, first premolar, or second premolar roots. Our study indicates that root volume assessment was not valuable in clinical decision-making on whether the cleft side canines would soon erupt.

Our study did not find statistically significant differences in the reciprocal mean root length ratio pre-operative or post-operatively in the first or the second premolar. Our study also did not find statistically significant differences in the reciprocal mean root length ratio pre-operatively in the canine. The mean pre-operative canine root length compared with an average canine root length in this study is 18.2% in the erupting group and 17.8% in the non-erupting group, which was not statistically different. This highlights that clinicians in our study were grafting before the conventionally recommended ½ to 2/3 of root development. This reflects the current preference of the Craniofacial team at this center to graft earlier to avoid potential ingress of the canine into the defect which would lead to probable failure of the graft. It is the experience of the team that canines will occasionally erupt with a proportionately lesser amount of root development so a more conservative approach is followed. Success rates of 94% at this center [27] validates this approach. However, long-term volumetric maintenance of the alveolar ridge may theoretically be enhanced by a more contemporaneous descent of the canine following the graft. Hence, the impetus to identify factors which may predict the eruptive phase of the canine with more precision.

Due to the skewness and kurtosis, we elected to perform a sensitivity analysis and then reran the independent sample *t*-tests and paired *t*-tests with a 2000 bootstrap sample. One difference was found between the paired sample *t*-test before and after the bootstrap examining the reciprocal of the mean root length ratio. In the second premolar in the non-erupting group, a bootstrap non-parametric test result was favored over the paired sample *t*-test result due to the data being skewed. The paired sample *t*-test found the second premolar in the non-erupting group to be significant, but after the bootstrap was performed and re-analyzed with the paired sample *t*-test, the *p*-value was not significant. However, the bootstrap paired sample *t*-test confidence intervals do not contain zero. Confidence intervals not containing zero indicated the result was statistically significant. In a small sample size, the confidence interval is more important than a point estimate from the *p*-value, where there might be little difference [28]. Therefore, we can conclude the second premolar in the non-erupting group is significant based on the confidence interval not containing zero from the bootstrap paired *t*-test analysis.

A limitation of our study is the small sample size due to the exclusion of many patients from the initial sample population due to patients with CBCTs pre- and post-operatively outside of the time range from the grafting procedure, patients with a syndrome that may affect dental eruption, failed bone grafts, and the presence of a cleft-side prime lateral necessitating early SABG. Additionally, another limitation of our study is the measurements for root volume and root length were performed manually. In future studies, using an automated measurement system [29] could be beneficial. Root morphology affects root volume of the tooth so evaluating root morphology could also be beneficial in future studies [30]. In future studies, it could be valuable to analyze the gingival shape of the alveolar cleft with intraoral scans [31]. Intraoral scans were not available for all the patients in our study; however, this could be an interesting future direction.

## 5. Conclusions

Volumetric assessment of root development from CBCTs was not a reliable factor in predicting a successful outcome for canine eruption following secondary alveolar grafting in patients with cleft lip and palate. In addition, conventional radiographic root length assessment in this group of patients did not offer better predictability either. Therefore, success for canine eruption following secondary alveolar grafting requires more clinical judgment than relying solely on root formation.

## Figures and Tables

**Figure 1 diagnostics-13-01642-f001:**
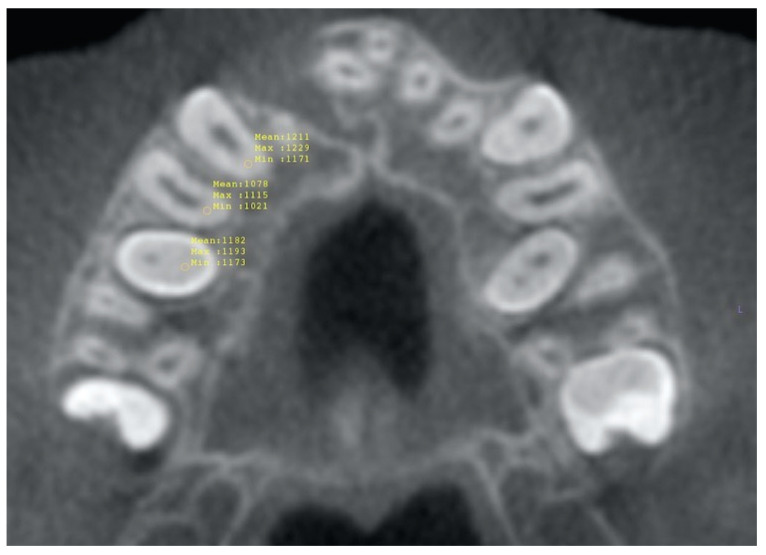
Root area was sectioned at the level of the CEJ and volumetric measurements (HU) were taken of the maxillary canine and premolar roots on the ipsilateral side with the alveolar cleft. This figure shows the volumetric measurements from a pre-graft CBCT.

**Figure 2 diagnostics-13-01642-f002:**
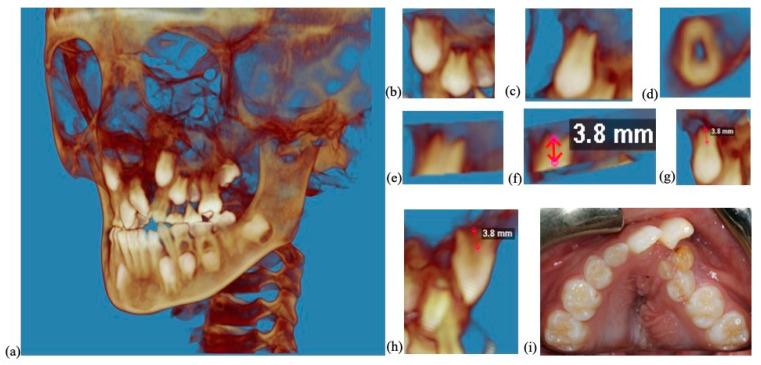
The roots of the maxillary canines and premolars on the ipsilateral side of the cleft were measured on pre-operative and post-operative CBCTs. This figure shows the root length measurement of a maxillary canine on a pre-operative CBCT. (**a**) Pre-operative CBCT of a patient with cleft lip and palate included in our study. (**b**) The canine and premolars on the ipsilateral side to the cleft were clipped from the CBCT. (**c**) The canine on the ipsilateral side to the cleft was clipped from the CBCT. (**d**) The root length viewed in the axial view. (**e**) The root was sectioned from the crown at the level of the buccal CEJ. (**f**) The root length was measured. (**g**,**h**) The root length measurement of the maxillary canine shown from different views. (**i**) Clinical photo showing an alveolar cleft.

**Table 1 diagnostics-13-01642-t001:** Mean root volume change (%) of the canine, first premolar and second premolar between pre- and post-operative CBCTs in erupting and non-erupting canine groups.

	Erupting Canine		Non-Erupting Canine		*p* Value
	Mean (%)	Standard Deviation (%)	Mean (%)	Standard Deviation (%)	
Canine	19.65	15.32	16.78	12.58	*NS*
1st Premolar	21.64	23.09	12.34	8.98	*NS*
2nd Premolar	21.68	26.82	17.17	12.83	*NS*

*NS* indicates a non-significant result (*p* value > 0.05).

**Table 2 diagnostics-13-01642-t002:** Pre-operative mean root length ratios for canine, first premolar and second premolar in erupting and non-erupting groups.

	Erupting Canine		Non-Erupting Canine		*p* Value
	Mean	Standard Deviation	Mean	Standard Deviation	
Canine	0.18	4.76	0.18	1.89	*NS*
1st Premolar	0.19	5.50	0.19	5.22	*NS*
2nd Premolar	0.12	8.41	0.11	8.87	*NS*

*NS* indicates a non-significant result (*p* value > 0.05).

**Table 3 diagnostics-13-01642-t003:** Post-operative mean root length ratios for canine, first premolar and second premolar in erupting and non-erupting groups.

	Erupting Canine		Non-Erupting Canine		*p* Value
	Mean	Standard Deviation	Mean	Standard Deviation	
Canine	0.43 *	0.77	0.35 *	0.86	0.025
1st Premolar	0.49	1.18	0.42	0.79	*NS*
2nd Premolar	0.33	1.43	0.29	1.72	*NS*

* indicates statistically significant difference (*p* value < 0.05).

**Table 4 diagnostics-13-01642-t004:** Paired t test results comparing pre- and post-operative reciprocal of mean root length ratios in the same patient for erupting and non-erupting groups.

	Erupting				Non-Erupting			
	Correlation	*p* Value Paired *t* Test	*p* Value Bootstrap Paired *t* Test	Power	Correlation	*p* Value Paired *t* Test	*p* Value Bootstrap Paired *t* Test	Power
**Canine**	0.53	<0.001 *	0.014 *	99%	0.48	<0.001 *	<0.001 *	100%
**1st** **Premolar**	0.76	<0.001 *	0.005 *	99%	0.35	<0.001 *	<0.001 *	100%
**2nd** **Premolar**	0.51	<0.001 *	0.033 *	95%	0.33	0.002 *	0.071	93%

* indicates significance at *p* value < 0.05.

## Data Availability

The data presented in this study are available upon request.

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
