# Peer review of "Evaluation of Dental Root Development Regarding Maxillary Canine Eruption Status after Secondary Alveolar Bone Grafting in Patients with Cleft Lip and Palate"

_diagnostics, 2023, doi:10.3390/diagnostics13091642_

Round 1

Reviewer 1 Report

Many thanks for the paper submission. This paper is very intersting, however some modifications are required in order to proceed.

1) at line 35 the authors should report a bref introduction regarding the concepts and materials for alveolar bone graft. Please add this phrase at the beginning of the paper

"... Alveolar bone grafting is a technique used for bone atrophy in order to restore an adequate bone volume: autologous, eterologous and xenograft are possible materials used in this technique.."

please consider citing the following:

Petretta M, Gambardella A, Boi M, Berni M, Cavallo C, Marchiori G, Maltarello MC, Bellucci D, Fini M, Baldini N, Grigolo B, Cannillo V. Composite Scaffolds for Bone Tissue Regeneration Based on PCL and Mg-Containing Bioactive Glasses. Biology (Basel). 2021 May 4;10(5):398. doi: 10.3390/biology10050398. PMID: 34064398; PMCID: PMC8147831.   Baldini N, De Sanctis M, Ferrari M. Deproteinized bovine bone in periodontal and implant surgery. Dent Mater. 2011 Jan;27(1):61-70. doi: 10.1016/j.dental.2010.10.017. Epub 2010 Nov 27. PMID: 21112618.   Chisci, G.; Hatia, A.; Chisci, E.; Chisci, D.; Gennaro, P.; Gabriele, G. Socket Preservation after Tooth Extraction: Particulate Autologous Bone vs. Deproteinized Bovine Bone. Bioengineering 2023, 10, 421. https://doi.org/10.3390/bioengineering10040421

2) please include in material and methods IRB approval, name of Institute and date of approval.

3) figure 2 is a very interesting image: could the authors include the same cbct with different slices?

4) at line 237 a brief introduction before the sentence "The null-hypothesis was accepted." should be included

Author Response

Dear reviewers,

We want to extend our deepest gratitude to all of you for providing valuable feedback and contributing to improving our manuscript.

Please see below for the itemized corrections to our revised manuscript.

Reviewer #1:

1) at line 35 the authors should report a bref introduction regarding the concepts and materials for alveolar bone graft. Please add this phrase at the beginning of the paper "... Alveolar bone grafting is a technique used for bone atrophy in order to restore an adequate bone volume: autologous, eterologous and xenograft are possible materials used in this technique.."

Thank you for your comment. We have briefly introduced the concept of alveolar bone grafting in cleft lip and palate patients and the materials used in the revised manuscript. However, we did not include the suggested sentence at the beginning of the manuscript because alveolar bone grafting in patients with cleft lip and palate is not used for bone atrophy as it is in a non-cleft patient. Instead, alveolar bone grafts in patients with cleft lip and palate bridge a discontinuous alveolus that congenitally never fused.

please consider citing the following:

Petretta M, Gambardella A, Boi M, Berni M, Cavallo C, Marchiori G, Maltarello MC, Bellucci D, Fini M, Baldini N, Grigolo B, Cannillo V. Composite Scaffolds for Bone Tissue Regeneration Based on PCL and Mg-Containing Bioactive Glasses. Biology (Basel). 2021 May 4;10(5):398. doi: 10.3390/biology10050398. PMID: 34064398; PMCID: PMC8147831.   Baldini N, De Sanctis M, Ferrari M. Deproteinized bovine bone in periodontal and implant surgery. Dent Mater. 2011 Jan;27(1):61-70. doi: 10.1016/j.dental.2010.10.017. Epub 2010 Nov 27. PMID: 21112618.   Chisci, G.; Hatia, A.; Chisci, E.; Chisci, D.; Gennaro, P.; Gabriele, G. Socket Preservation after Tooth Extraction: Particulate Autologous Bone vs. Deproteinized Bovine Bone. Bioengineering 2023, 10, 421. https://doi.org/10.3390/bioengineering10040421

Thank you for the suggested articles. We appreciate the time you put in to help support our manuscript. These articles discuss the grafts that should be used in periodontal procedures, implants, and edentulous spaces. Therefore, it would not be pertinent to our study to cite them for alveolar bone grafting in patients with cleft lip and palate. However, the authors acknowledge the need to cite articles discussing alveolar bone grafting materials in patients with cleft lip and palate and have added references in the revised manuscript.

2) please include in material and methods IRB approval, name of Institute and date of approval.

Thank you for your comment. This information has been added to the revised manuscript.

3) figure 2 is a very interesting image: could the authors include the same cbct with different slices?

Thank you for your suggestion. We have updated Figure 2 to have more slices from the CBCT and updated the caption.

4) at line 237 a brief introduction before the sentence "The null-hypothesis was accepted." should be included

Thank you for your comment. This has been corrected in the revised manuscript.

Reviewer 2 Report

Dear Authors, first of all, congratulations for the study. The sample size is impressive. 

However during the manuscript reading I have found some points to be amended, both for understanding and study replication as well as literature comparison. Therefore I recomend Major revisions.

In particular: 

When You mention that patients affected by Cleft Palate are subjected to tooth development alteration, You should specify which ones, and based on which pathogenetic mechanism. 

Materials and Methods, 

I did not understand why You used two different softweres for measurments. This is a limit of the study. 

I did not fully understand the bootstraps test, and You should add a summarizing sentence so that who is not expert in statistics can fully understand what did You do and why. In addition the softwere with the version  should be better specified. 

Discussion. 

you did well in presenting Your results but You did not discuss with data in literature in terms of morpology and anatomy. As regards the root volume I reccomend this article to understand the limitation of Your study in the measurments and also in not evaluated the crown/root lenght as well as the used methods (Di Angelo L, Di Stefano P, Bernardi S, Continenza MA. A new computational method for automatic dental measurement: The case of maxillary central incisor. Comput Biol Med. 2016 Mar 1;70:202-209. doi: 10.1016/j.compbiomed.2016.01.018. Epub 2016 Jan 22. PMID: 26851728.; Kuralt M, Cmok Kučič A, Gašperšič R, Grošelj J, Knez M, Fidler A. Gingival shape analysis using surface curvature estimation of the intraoral scans. BMC Oral Health. 2022 Jul 12;22(1):283. doi: 10.1186/s12903-022-02322-y. Erratum in: BMC Oral Health. 2022 Dec 9;22(1):577. PMID: 35820843; PMCID: PMC9275066.; Haberthür D, Hlushchuk R, Wolf TG. Automated segmentation and description of the internal morphology of human permanent teeth by means of micro-CT. BMC Oral Health. 2021 Apr 12;21(1):185. doi: 10.1186/s12903-021-01551-x. PMID: 33845806; PMCID: PMC8040229.) 

finally English needs and extensive revisions. I usually do not comment about English but this time I have found difficulties in understanding also the conclusion of Your work. 

Author Response

Dear reviewers,

We want to extend our deepest gratitude to all of you for providing valuable feedback and contributing to improving our manuscript.

Please see below for the itemized corrections to our revised manuscript.

Reviewer #2:

When You mention that patients affected by Cleft Palate are subjected to tooth development alteration, You should specify which ones, and based on which pathogenetic mechanism.

Thank you for your suggestion. We have added a discussion about delayed tooth development and pathogenetic mechanism in children with cleft lip and palate to the revised manuscript.

Materials and Methods, 

I did not understand why You used two different softweres for measurments. This is a limit of the study. 

Thank you for your comment. We decided to use two different software programs for root length measurement and root volume assessment because each software was more specialized in its respective analysis. Therefore, the two measurements, root length, and root volume were not compared to each other in our study.

I did not fully understand the bootstraps test, and You should add a summarizing sentence so that who is not expert in statistics can fully understand what did You do and why. In addition the softwere with the version  should be better specified. 

Thank you for your suggestion. We have added an expanded explanation of bootstrapping, including what it is and why we performed this test, to the revised manuscript. Additionally, we also added the software version to the revised manuscript.

Discussion. 

you did well in presenting Your results but You did not discuss with data in literature in terms of morpology and anatomy.

Thank you for your comment. Our study did not look at the morphology or anatomy of the roots. We limited our analysis to the root length and root volume. However, the morphology or anatomy of the roots would be an interesting direction to explore in a follow-up study.

As regards the root volume I reccomend this article to understand the limitation of Your study in the measurments and also in not evaluated the crown/root lenght as well as the used methods (Di Angelo L, Di Stefano P, Bernardi S, Continenza MA. A new computational method for automatic dental measurement: The case of maxillary central incisor. Comput Biol Med. 2016 Mar 1;70:202-209. doi: 10.1016/j.compbiomed.2016.01.018. Epub 2016 Jan 22. PMID: 26851728.; Kuralt M, Cmok Kučič A, Gašperšič R, Grošelj J, Knez M, Fidler A.

Thank you for your comment. We agree that having a computational method for dental measurement would be beneficial, and the authors acknowledge this as a limitation of the study as the teeth were measured manually. Additionally, as you mentioned, we did not calculate the crown/root when investigating the root development. We chose only to evaluate the root development and not the crown/root ratio due to the variable nature of clinical crowns in patients with cleft lip and palate. We decided to use only the root so that if the crown was morphologically different from the norms, this study could still be clinically valuable for clinicians on craniofacial teams-- this is a custom protocol in the dental literature. The root development is more important than the crown/root ratio for determining dental development.

Gingival shape analysis using surface curvature estimation of the intraoral scans. BMC Oral Health. 2022 Jul 12;22(1):283. doi: 10.1186/s12903-022-02322-y. Erratum in: BMC Oral Health. 2022 Dec 9;22(1):577. PMID: 35820843; PMCID: PMC9275066.; 

Thank you for sharing this article with us. Although this would have been very interesting for our study, we used historical data from existing CBCT scans. Also, the cleft center did not take intraoral or optical scans of these patients. The CBCTs are radiographs, and gingival shape/soft tissue cannot be accessed. The authors acknowledge that if intraoral scans were also available for this data set, this would be an exciting addition to the study.

Haberthür D, Hlushchuk R, Wolf TG. Automated segmentation and description of the internal morphology of human permanent teeth by means of micro-CT. BMC Oral Health. 2021 Apr 12;21(1):185. doi: 10.1186/s12903-021-01551-x. PMID: 33845806; PMCID: PMC8040229.) 

Thank you for the recommendation. This was an interesting article; however, this method could not be applied to our study as teeth used in micro-CT analysis are extracted as this is an ex vivo technique to our knowledge.

Finally English needs and extensive revisions. I usually do not comment about English but this time I have found difficulties in understanding also the conclusion of Your work. 

Thank you for your recommendation. We have revised our article's English and improved the language in conclusion for clarity.

Reviewer 3 Report

Dear Authors,

thank you for this interesting paper. There are some suggestions of mine how to improve it though:

1. Please, delete the scientiffic degrees next to the authors' names.

2. Lines 41-43, please explain what you mean by alveolar bone grafting in infants?

3. The authors should add the information, what exactly do they mean by "cleft" - there are different types of them. As far as I noticed, the isolated cleft palate and isolated cleft lip were not taken into account. What about the other types? There should be a short note on the division of cefts in general, in the introduction. Besides, I think you should add the sequences of bone grafting in the introduction (focus on primary bone grafting - stages and leave the note about the secondary one; maybe a table or a figure would be helpful?) and the possible donor side (ilac crest), eg.

Paradowska-Stolarz, A.; Mikulewicz, M.; Duś-Ilnicka, I. Current Concepts and Challenges in the Treatment of Cleft Lip and Palate Patients—A Comprehensive Review. J. Pers. Med. 202212, 2089. https://doi.org/10.3390/jpm12122089

4. Please, add the information on the healing of the three types of grafts - does the primary graft heal quicker? Is it more effective? If so, why?

5. Line 104-106 should be rather incorporated into introduction, not M&M's

6. Line 145 - I would rather use "statistical analysis"

7. Lines 169-171 refer more to M&M section

8. Line 238 - I would incorporate the sentence about the null hypothesis into the further sentences.

9. I would also incorporate to the discussion the possible problems that may occur due to the soft tissues surrounding the clefted alveolus, called Simonart's band, eg. 

- Ariawan D, Vitria EE, Sulistyani LD, et al. Prevalence of Simonart’s band in cleft children at a cleft center in Indonesia: A nine-year retrospective study. Dent Med Probl. 2022;59(4):509–515. doi:10.17219/dmp/145065

10. The references are pretty old ones, please incorporate new ones (from the past 5 years) and extend its number to 25 as minimum - there are plenty of researches on the bone grafting.

To sum up, the article needs extensive revisions before publishing it. 

Author Response

Dear reviewers,

We want to extend our deepest gratitude to all of you for providing valuable feedback and contributing to improving our manuscript.

Please see below for the itemized corrections to our revised manuscript.

Reviewer #3:

  1. Please, delete the scientiffic degrees next to the authors' names.

Thank you for your suggestion. We have corrected this in the revised manuscript.

  1. Lines 41-43, please explain what you mean by alveolar bone grafting in infants?

Thank you for your comment. For clarity, we have revised this sentence to be children who underwent primary bone grafting for clarity. Infants (aged 0-24 months) can receive primary alveolar bone grafting.

  1. The authors should add the information, what exactly do they mean by "cleft" - there are different types of them. As far as I noticed, the isolated cleft palate and isolated cleft lip were not taken into account. What about the other types? There should be a short note on the division of cefts in general, in the introduction. Besides, I think you should add the sequences of bone grafting in the introduction (focus on primary bone grafting - stages and leave the note about the secondary one; maybe a table or a figure would be helpful?) and the possible donor side (ilac crest), eg.

Thank you for your comments. We have addressed all of your suggestions in our revised manuscript. 1) The authors have added a note about cleft lip and/or palate divisions in the revised manuscript. 2) In the revised manuscript, we have specified the methods by which cleft (unilateral cleft lip and palate) was included in this study. 3) We have added information about the sequencing of primary bone grafting to the revised manuscript. However, primary bone grafting was included simply for historical context and completeness when discussing the classifications. Still, no center we know of currently performs alveolar bone grafting due to its harmful effects on maxillary growth. 4) We have included information about the donor site in the revised manuscript.

- Paradowska-Stolarz, A.; Mikulewicz, M.; Duś-Ilnicka, I. Current Concepts and Challenges in the Treatment of Cleft Lip and Palate Patients—A Comprehensive Review. J. Pers. Med. 2022, 12, 2089. https://doi.org/10.3390/jpm12122089

Thank you for sharing this article with us. We have added this reference and cited it in the revised manuscript.

  1. Please, add the information on the healing of the three types of grafts - does the primary graft heal quicker? Is it more effective? If so, why?

Thank you for your comment. The primary, secondary, and tertiary bone grafts are similar in healing. The distinction is based on when the graft is performed, as stated in lines 53-73. We are unaware of primary grafts being performed at any cleft centers currently, and primary alveolar bone grafting was briefly discussed for completeness when introducing the timing of alveolar grafting.

  1. Line 104-106 should be rather incorporated into introduction, not M&M's

Thank you for your suggestion. We have moved these lines to the introduction in the revised manuscript.

  1. Line 145 - I would rather use "statistical analysis"

Thank you for your comment. We have corrected this in the revised manuscript.

  1. Lines 169-171 refer more to M&M section

Thank you for your suggestion. We have removed these lines from the results and moved them to the methods section in the revised manuscript.

  1. Line 238 - I would incorporate the sentence about the null hypothesis into the further sentences.

Thank you for your suggestion. This has been corrected in the revised manuscript.

  1. I would also incorporate to the discussion the possible problems that may occur due to the soft tissues surrounding the clefted alveolus, called Simonart's band, eg. 

- Ariawan D, Vitria EE, Sulistyani LD, et al. Prevalence of Simonart’s band in cleft children at a cleft center in Indonesia: A nine-year retrospective study. Dent Med Probl. 2022;59(4):509–515. doi:10.17219/dmp/145065

Thank you for your suggestion and for sharing this article. Simonart’s bands indicate a soft tissue bridge between the boney segments, which is included in the diagnosis of incomplete cleft lip and palate. However, our study only included complete cleft lip and palate patients, and as a result, Simonart’s band is not applicable to our research, and this is why we have not included this in our discussion.

  1. The references are pretty old ones, please incorporate new ones (from the past 5 years) and extend its number to 25 as minimum - there are plenty of researches on the bone grafting.

Thank you for your recommendation. We have incorporated references from the past five years and increased the number of references to 25 in the revised manuscript.

3) figure 2 is a very interesting image: could the authors include the same cbct with different slices?

Thank you for your suggestion. We have updated Figure 2 to have more slices from the CBCT and updated the caption.

4) at line 237 a brief introduction before the sentence "The null-hypothesis was accepted." should be included

Thank you for your comment. This has been corrected in the revised manuscript.

Round 2

Reviewer 1 Report

Dear authors, many thanks for the modifications of your paper. However, some major flaws lacks for a proper article for the Diagnostic journal.

1) at the previous revision the authors rejected the invited modifications: the motivations reported in the point by point are not sufficient, so I'm asking you to reconsider the modifications requested. Here you have another occasion to modify your paper:

at line 53 the authors should report a bref introduction regarding the concepts and materials for alveolar bone graft. Please add this phrase at the beginning of the paper "... Alveolar bone grafting is a technique used for bone atrophy in order to restore an adequate bone volume: autologous, eterologous and xenograft are possible materials used in this technique.."

please consider citing the following:

Petretta M, Gambardella A, Boi M, Berni M, Cavallo C, Marchiori G, Maltarello MC, Bellucci D, Fini M, Baldini N, Grigolo B, Cannillo V. Composite Scaffolds for Bone Tissue Regeneration Based on PCL and Mg-Containing Bioactive Glasses. Biology (Basel). 2021 May 4;10(5):398. doi: 10.3390/biology10050398. PMID: 34064398; PMCID: PMC8147831. Baldini N, De Sanctis M, Ferrari M. Deproteinized bovine bone in periodontal and implant surgery. Dent Mater. 2011 Jan;27(1):61-70. doi: 10.1016/j.dental.2010.10.017. Epub 2010 Nov 27. PMID: 21112618. Chisci, G.; Hatia, A.; Chisci, E.; Chisci, D.; Gennaro, P.; Gabriele, G. Socket Preservation after Tooth Extraction: Particulate Autologous Bone vs. Deproteinized Bovine Bone. Bioengineering 2023, 10, 421. https://doi.org/10.3390/bioengineering10040421

2) The name of the Institute that approved this research still lacks in the methods: this information is required, as the readers could suspect the nonexistence of an ethical evaluation and this article could create ethical difficulties for the journal.

3) a new figure with clinical image of the presented graft should be added.

4) at line 221 please add the information regarding the software that your personel used for this research.

5) a better introduction before the report "the null hypotesis was accepted" should be introducted at the beginning of the discussion section.

Author Response

Dear Reviewer,

Thank you for your suggestions to improve our manuscript. We appreciate your feedback. Please see below for the itemized list for each revision.

Reviewer #1:

1) at the previous revision the authors rejected the invited modifications: the motivations reported in the point by point are not sufficient, so I'm asking you to reconsider the modifications requested. Here you have another occasion to modify your paper:

at line 53 the authors should report a bref introduction regarding the concepts and materials for alveolar bone graft. Please add this phrase at the beginning of the paper "... Alveolar bone grafting is a technique used for bone atrophy in order to restore an adequate bone volume: autologous, eterologous and xenograft are possible materials used in this technique."

Thank you for your comment. We have added the suggested phrase to our manuscript (at line 48 in the second revision of the manuscript which is line 53 in the original manuscript). The materials for alveolar bone grafting are discussed in lines 73-87. The concepts of alveolar bone grafting are discussed in lines 50-72.

please consider citing the following:

Petretta M, Gambardella A, Boi M, Berni M, Cavallo C, Marchiori G, Maltarello MC, Bellucci D, Fini M, Baldini N, Grigolo B, Cannillo V. Composite Scaffolds for Bone Tissue Regeneration Based on PCL and Mg-Containing Bioactive Glasses. Biology (Basel). 2021 May 4;10(5):398. doi: 10.3390/biology10050398. PMID: 34064398; PMCID: PMC8147831. Baldini N, De Sanctis M, Ferrari M. Deproteinized bovine bone in periodontal and implant surgery. Dent Mater. 2011 Jan;27(1):61-70. doi: 10.1016/j.dental.2010.10.017. Epub 2010 Nov 27. PMID: 21112618. Chisci, G.; Hatia, A.; Chisci, E.; Chisci, D.; Gennaro, P.; Gabriele, G. Socket Preservation after Tooth Extraction: Particulate Autologous Bone vs. Deproteinized Bovine Bone. Bioengineering 2023, 10, 421. https://doi.org/10.3390/bioengineering10040421

Thank you for the suggested articles. We have added all of these suggested articles to our paper.

2) The name of the Institute that approved this research still lacks in the methods: this information is required, as the readers could suspect the nonexistence of an ethical evaluation and this article could create ethical difficulties for the journal.

Thank you for your comment. The institution that approved the research was Boston Children’s Hospital. We are very sorry for the confusion and we re-worded sentence 141 to improve clarity.

3) a new figure with clinical image of the presented graft should be added.

Thank you for your suggestion. We have updated the figure to include a clinical photo.

4) at line 221 please add the information regarding the software that your personel used for this research.

Thank you for your comment. We have added the information regarding the statistics software.

5) a better introduction before the report "the null hypothesis was accepted" should be introduced at the beginning of the discussion section.

Thank you for your comment. We have revised the introduction before the “null hypothesis was accepted” to improve this area of our manuscript.

Reviewer 2 Report

Dear Authors discussion has not been improved according go the suggestione  please discuss your data and your methods with literature provided. 

Author Response

Dear Reviewer,

Thank you for your suggestions to improve our manuscript. We appreciate your feedback. Please see below for the itemized list for each revision.

Reviewer #2:

Dear Authors discussion has not been improved according to the suggestion please discuss your data and your methods with literature provided. 

Thank you for your comment. We have revised the discussion according to your suggestions and discussed the data in terms of limitations of our methods with the literature provided in the first round of revisions.

Reviewer 3 Report

Thank you for the changes and explanations to my comments. In this for the paper could be accepted. Best regards!

Author Response

Dear Reviewer,

Thank you for your suggestions to improve our manuscript. We appreciate your feedback. Please see below for the itemized list for each revision.

Reviewer #3:

Thank you for the changes and explanations to my comments. In this for the paper could be accepted. Best regards!

Thank you for your suggestions to help improve our manuscript!

Round 3

Reviewer 1 Report

yes